# Malaria Transmission in Sahelian African Regions, a Witness of Climate Changes

**DOI:** 10.3390/ijerph191610105

**Published:** 2022-08-16

**Authors:** Ronan Jambou, Medard Njedanoun, Geremy Panthou, Luc Descroix

**Affiliations:** 1Centre de Recherche Médicale et Sanitaire (CERMES), Niamey BP 10887, Niger; 2Global Health Department, Institut Pasteur, 75015 Paris, France; 3UMR 5564, Université Grenoble Alpes (UGA), IRD/CNRS, BP 53, CEDEX 09, 38041 Grenoble, France; 4IRD, UMR PALOC, IRD/Museum National Histoire Naturelle, CEDEX 05, 75231 Paris, France

**Keywords:** climate change, floods, malaria, Sahel

## Abstract

Climate changes in the eastern part of Sahelian regions will induce an increase in rainfalls and extreme climate events. In this area, due to the intense events and floods, malaria transmission, a climate sensitive disease, is thus slowly extending in time to the drought season and in areas close to the border of the desert. Vectors can as well modify their area of breeding. Control programs must be aware of these changes to adapt their strategies.

## 1. Introduction

West Africa is identified as a hot spot for climate change impact because of intense earth-atmosphere interactions and higher temperature [1]. In Africa, temperature increases 1.5 faster than the global temperature. Predictive models confirm an increase of rainfall in the eastern and the central part of the Sahel from Niger to Ethiopia whereas drought will increase after 2030 in the western portion of West Africa from Mali to Mauritania (see [1] for review). One of the most important issues of the next decades will thus be the impact of these climate changes on the transmission of vector-borne diseases, including transmission of malaria. 29 out of the 85 countries experiencing malaria account for 96% of the whole cases and deaths from the disease. However, 55% of cases are declared by six Africa countries, Nigeria (26.8%), the Democratic Republic of the Congo (12.0%), Uganda (5.4%), Mozambique (4.2%), Angola (3.4%), and Burkina Faso (3.4%) [2]. Despite their large dry Sahelian region, Niger and Burkina Faso are in the top ten of these countries declaring malaria cases.

## 2. In Sahel, by the River, Malaria Transmission Is Extending to the Dry Season after Floods

In West Africa, malaria transmission usually occurs over six months from June to December, linked to the rainy season. It peaks from August to September, aligning with the beginning of the rainy season. Moreover, the mosquito burden is very variable depending on the availability of surface water and breeding sites. This risk of transmission can thus range from 50 to 250 bites/human/year in the South to less than 20 in the North of the Sahelian area [3]. Nevertheless, in Niger, the Saharan area covers more than 55% of the surface of the country. However, malaria is accounting for 28% of all the diseases occurring under five years and 50% of the deaths among children. High levels of malaria transmission are concentrated in the south part of the country, with very efficient malaria vectors as *An. funestus*, *An. gambiae ss* and *An. arabiensis*. Other secondary species are also described, such as *An. moucheti* and *An. nili* [4]. In 2020–2021, according to the National Malaria Control Program (NMCP), an increase in malaria cases of 30% has been recorded in Niger in comparison with 2019 (2.19, 2.85, 2.99 millions of cases in 2019, 2020, 2021 respectively). Concomitantly, malaria transmission has extended during a longer period, covering a large part of the drought season up to February and March (Malaria National Control Program, report 2021).

Despite the impact of COVID-19 on the disruption of health services and on malaria cases, during the period when malaria transmission was increasing, floods have been observed in Niger, especially in Niamey and in Mali in 2019 and 2020. They were due to flooding of the Niger river [5]. This flood was itself related to a modification of the double seasonal flood of the river with an increase of the first one (August–September) associated with a change in the Sahelian climatic conditions during the last decades. These changes correlate with an increase of storms in this area [6].

In the Middle Niger River Valley (MNRV), two floods are yearly observed, (i) the local red one owing to the streams coming from local rivers from July to September, and (ii) the Guinean black flood originating in the Fouta Djallon, from October to January (Figure 1A). The Guinean flood used to be the principal event, but since 1984 the red flood peak exceeded the black flood, and this red flood peak increase from 1000 m^3^/s to 1500 m^3^/s (Figure 1A) since 2011 [7]. This was reinforced by an increase in rainfall intensity and in the frequency of rainfall extreme events. This increase in runoff observed during and after the drought caused an increase in the number, in the size, in the volume and in the duration of seasonal ponds, some of them becoming permanent, inducing a paradoxical rise in groundwater in sedimentary areas [8].

The first Sahelian hydrological paradox appeared during the long 20th century drought (1968–1993), when an increase in runoff coefficients and of streams was observed. The second one is noticed since the very end of the 20th century and the rainfall recovery. It is still not clear if these changes are linked to fluctuation of the Sahelian climate, to global warming, or both. However, this general regreening of the Sahel was not associated with a decrease in runoff as expected. Moreover, in towns, informal urbanization disrupts natural run-offs, increasing floods.

Overall, these processes deeply impact vector-transmitted disease. Indeed, as most of the Sahelian sedimentary areas are endorheic ones, this has led to the appearance of seasonal, and permanent ponds (Figure 1B), which are highly suitable as *Anopheles* breeding sites. However, unlike proliferation of *Aedes* (and transmission of arbovirus), period of runoffs and streams are usually not suitable for *Anopheles* outbreaks (and malaria transmission) as their larva are drowned.

During the second part of the rainy season, rainfalls occurring in Guinea are filling again the Niger river when the previous floods are still present, inducing very large pools and ponds. When the river discharges, these ponds overwhelmed urban rice fields, becoming suitable *An. gambiae* breeding sites.

These changes in the inter-annual rainfall regiment are not new and their origins are unclear. Indeed, the Sahelian rainfall regime is characterized by a very high spatio-temporal variability of rainfall, a strong seasonal cycle (Figure 2A) and a North-South gradient in rainfalls. Rainfalls are due exclusively to convective processes, generating very intense events lasting only a few hours. This monsoon system has a very high interannual to decadal variability [9]. Since the 1950s, the evolution of the Sahelian rainfall pattern can be summarized in three periods, (i) a wet phase (the 1950s and 1960s), followed by (ii) the “great Sahelian drought” (1970 through to the mid-1990s), and (iii) a recent period of hydro-climatic intensification (Figure 2B). This “great Sahelian drought” affected the entire Sahelian region for more than 20 years. To date, rainfall remains rare as the number of rainy days remains close to that observed during the “great drought” (Figure 2C) [10]. Ongoing works (CMIP6 simulations) suggest that this new hydro-climatic era is attributable in part to global warming (Figure 2D) [1,10,11,12] but also to internal variability [13,14]. Overall, rainfalls in Sahel re-increased to their former level of the 1960s but with a lower number of rainy days. This induces an increase in intense rainfall events, and higher flood hazard. This recent intensification of rainfalls seems to corelate with an extension to the North of areas submitted to mosquito-borne diseases. This hydro-climatic intensification is expected to continue in the coming decades. It can be thus expected that the current trend of extension of malaria transmission in the Saharan fringes will last in the same line.

During the same time, these rainfalls let water stream to tidal flats, leading to more or less permanent ponds and floods. At the end of the rainy season and during the drought, large areas of surface water remain. During low-flow, large ponds can also be observed near the rivers as rest of flooding. Concomitantly, surface water can also be observed as a result of overfilled groundwaters, cropping out at the surface (seen at Agadez, Niamey etc.). These marsh lands offer numerous breeding habitats suitable for *An. gambiae s.s*, *An. funestus*, and *An. nili* as observed, during the previous years, in the neighborhoods of large towns like Niamey and Maradi. Indeed, in 2020, in Niamey, very large floods invaded the town and wet area, persisting by the river up to the end of December. Along the river, the number of *Anopheles* collected during drought was greater than during the rainy season, and the number of bites per night increased also during cold season, supported by night temperature higher than usual. During this time the peak of malaria cases shifted to the week 40th (Figure 1C,D). This was more potent in dispensaries closed to the river (Figure 1D) than in those far from the water (Figure 1C). This trend will continue over the whole century, and intense rainfall events can be, in Sahelian area, key drivers of changes in malaria transmission [9].

## 3. Adaptation of the Malaria Control Activities Is Now Needed

Data related to malaria collected in Niger during the last years suggest an evolving situation in time and space of the transmission. These changes must be taken into account by the malaria control programs. However, the last period was heavily affected by the COVID-19 pandemic, which could have impacted the registration of cases. In the same vain, if most of climate models argue for an increase in rainfall in Sahel, the scale of the data generated does not for allow a fine prevision of the local impact of these changes and lot of works remain to be done. The objective of this short manuscript was to support the urgent need of new studies in this field.

Current strategies of malaria control in the Sahelian area, consist mainly of impregned bednets, treatment of symptomatic cases and seasonal malaria chemoprevention (SMC) [15]. For regions with more than 600 mm rainfall per year, four cycles of SMC are applied to all children under 5 years to prevent malaria attacks [15]. This last strategy is conducted mostly over the rainy season from August to November. In Niger 4.5 million children from 0 to 5 years have been monthly treated and 18 million treatments have been delivered. However, changes observed in the transmission of the disease imply modifications of this strategy. In area where remaining ponds exist, or on the edge of the rivers, malaria transmission extends to January to March, and additional SMC rounds should be considered in January or February to prevent malaria and child mortality.

At the same time control activities should be extended to the North, closer to the Saharan area, where an increase of up to 90% of malaria cases has been registered during the last two years (2020–2021). Extension of the area of *An. funestus* can be expected as well, which will support deployment of indoor house insecticide spreading strategy in Niger

## 4. Conclusions

Climate changes associated with more frequent intense rainfall will modify geographic and seasonal occurrence of malaria transmission in the Sahel region. A new round of studies should be conducted to adapt the control strategies in these countries and to anticipate new situations, like transmission in the desert fringes or extension of *An. funestus* and *An. stephensi* areas in the Sahelian stripe.

## Figures and Tables

**Figure 1 ijerph-19-10105-f001:**
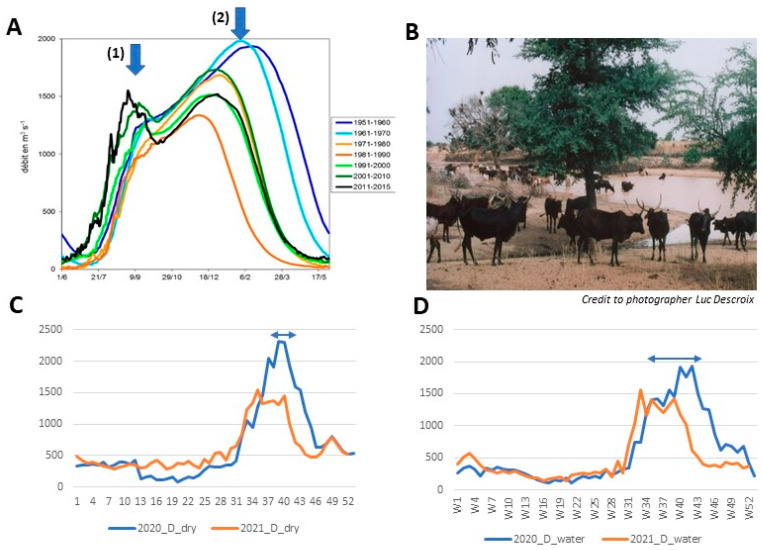
Change over the last 50 years of the two main floods of the Niger river in Niamey and its impact on malaria transmission. (**A**) Decadal evolution of the two floods of Middle Niger river at Niamey gauge station since 1950: (1) first flood due to rainfalls in Niger, (2) second flood due to rainfalls in Guinea. Over time the first pic increases. (**B**) In July, a pond in the Lullemeden sedimentary basin in Niger. Credit to photographer Luc Descroix. (**C**) Confirmed malaria cases registered in dispensaries from Niamey, far from the river. (**D**) Confirmed malaria cases registered in dispensaries closed to the river. Arrows highlight the shift of the transmission between a year with flooding (2020) and a year without flooding (2021). Raw number of cases are recorded and should be related to the population of each area. However, these graphs aim to highlight the shift in the period.

**Figure 2 ijerph-19-10105-f002:**
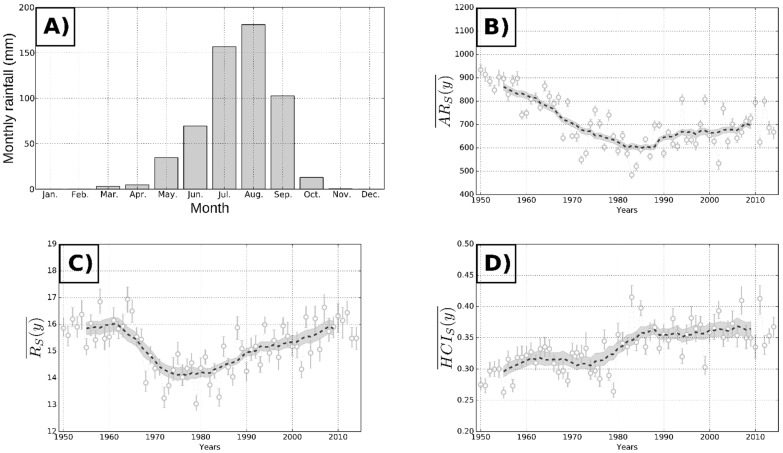
Evolution of rainfalls in the Sahel since 1950s, as registered in Niamey station. (**A**) Monthly rainfall at the Niamey (Niger) station with a seasonal effect. (**B**) Annual rainfall amounts in millimeters. (**C**) Mean intensity of wet days (average millimeters of water fallen during the rainy days), showing an increase since 2000. (**D**) Hydro-climatic intensity index, which measures the frequency of intense events, showing a progressive increase since 1990.

## Data Availability

Raw data are available from the authors under reasonable demands.

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
