# Peer review of "Malaria Transmission in Sahelian African Regions, a Witness of Climate Changes"

_ijerph, 2022, doi:10.3390/ijerph191610105_

Round 1

Reviewer 1 Report

The subject of malaria transmission in Sahelian countries is a very important topic and the reasons for changes are of primary importance for better controlling this disease. The role of climatic changes and flooding on mosquito production are key parameters in relation with the malaria transmission. However, a lot of data are missing in the current manuscript to support the findings. More references are needed and the conclusion must take into account the overall factors and parameters, including the access to health services, which were hugely disrupted during the COVID-19 pandemic. Some more detailed questions are included into the attached file and must be answered to provide a more exhaustive understanding of the discussion and findings.

Author Response

to follow the reviewer , we add our reply in the track pdf following his own questions

Reviewer 2 Report

(ijerph-1814190-peer-review-comments v1)

Brief Report

Malaria transmission in Sahelian African regions, a witness of climate changes

Ronan Jambou, Medard Njedanoun, Geremy Panthou and Luc Descroix

 1.      Recommendation:

     Accept after minor revisions.

 2.      Comments to the author:

 The brief note is based on the information related to climate change, hydrology, and some malaria predicting models in the Sahel. Based on these studies and the observation of malaria case increase by 40% in the areas by the villages in Sahel, the authors have drawn some conclusions related to malaria transmission and change in strategies for SMC, treatment of symptomatic cases, and insecticide spraying strategy. But the authors have not presented the supporting malaria data of these villages to substantiate the facts. The figures and graphs presented in the manuscript are not clear. Also, the authors have not discussed prevailing malaria and vector control strategies in these areas. The note points out the extension of malaria transmission comparing 2020 and 2021.

Malaria epidemiology is complex and involves a number of parameters like the required humidity, and temperature over a period of time to allow the parasite development inside the mosquito and transmission to the human host, vector density and longevity, etc. However, since the authors have already suggested that a new round of studies should be conducted to adapt the control strategies in these countries and to anticipate new situations, this brief note can be considered for publication as an observational note after suggested changes.  

 Specific comments: 

Abstract:

Line no. 25-28: It would be appropriate to revise line no. 25-28 in the abstract on the lines of the data presented in the World Malaria Report (2021).  The WMR (2021) reads  “In 2020, 29 of the 85 countries that were malaria endemic (including the territory of French Guiana) accounted for about 96% of malaria cases and deaths globally. Nigeria (26.8%), the Democratic Republic of the Congo (12.0%), Uganda (5.4%), Mozambique (4.2%), Angola (3.4%), and Burkina Faso (3.4%) accounted for 55% of all cases. Four countries accounted for just over half of all malaria deaths globally: Nigeria (31.9%), the Democratic Republic of the Congo (13.2%), the United Republic of Tanzania (4.1%), and Mozambique (3.8%)”.

 Line 122: ‘Pic of malaria cases shifted to’—needs correction. Probably, the authors mean to say ‘peak of malaria season’

 Lines 140 & 141: There is nothing like insecticide spreading strategy in malaria. Please check.

Figure 1: A, C, D: X & Y axis are not clear.

Figure 2.:  figures are blurred and not clear.

                  It is suggested to replace them with legible figures. 

Ethics approval and consent to participate: It appears data used for this study had approval by the National, ethics committee of Niger for a different purpose. Would the same ethical clearance hold good for this publication also?  

Author Response

Reviewer 2

Comments and Suggestions for Authors

The subject of malaria transmission in Sahelian countries is a very important topic and the reasons for changes are of primary importance for better controlling this disease. The role of climatic changes and flooding on mosquito production are key parameters in relation with the malaria transmission. However, a lot of data are missing in the current manuscript to support the findings. More references are needed and the conclusion must take into account the overall factors and parameters, including the access to health services, which were hugely disrupted during the COVID-19 pandemic. Some more detailed questions are included into the attached file and must be answered to provide a more exhaustive understanding of the discussion and findings

>> this text is a discussion on some aspects of floods , and is not attempting to bring all data about the problem but to show parallele between climatic data and malaria data . The propblem encouted now adays is not always the variation of the number of malaria attacks, but the the modifications of their time and spacial repartitions which impact the stratgeis to apply. Papers presenting original data from the villages are in process.

Comments to the author:

The brief note is based on the information related to climate change, hydrology, and some malaria predicting models in the Sahel. Based on these studies and the observation of malaria case increase by 40% in the areas by the villages in Sahel, the authors have drawn some conclusions related to malaria transmission and change in strategies for SMC, treatment of symptomatic cases, and insecticide spraying strategy. But the authors have not presented the supporting malaria data of these villages to substantiate the facts. The figures and graphs presented in the manuscript are not clear. Also, the authors have not discussed prevailing malaria and vector control strategies in these areas. The note points out the extension of malaria transmission comparing 2020 and 2021.

>> this text is a discussion on some aspects of floods , it is not attempting to bring all data about the problem but to show parallels between climatic data and malaria data . Papers presenting original data from the village are in process. 

Malaria epidemiology is complex and involves a number of parameters like the required humidity, and temperature over a period of time to allow the parasite development inside the mosquito and transmission to the human host, vector density and longevity, etc. However, since the authors have already suggested that a new round of studies should be conducted to adapt the control strategies in these countries and to anticipate new situations, this brief note can be considered for publication as an observational note after suggested changes. 

>> the main target of this note is the financial partners and researcher which must be aware for the need of new studies as all what we can see on TV speak about dryness etc. which is not really true.

Specific comments:

Abstract:

Line no. 25-28: It would be appropriate to revise line no. 25-28 in the abstract on the lines of the data presented in the World Malaria Report (2021).  The WMR (2021) reads  “In 2020, 29 of the 85 countries that were malaria endemic (including the territory of French Guiana) accounted for about 96% of malaria cases and deaths globally. Nigeria (26.8%), the Democratic Republic of the Congo (12.0%), Uganda (5.4%), Mozambique (4.2%), Angola (3.4%), and Burkina Faso (3.4%) accounted for 55% of all cases. Four countries accounted for just over half of all malaria deaths globally: Nigeria (31.9%), the Democratic Republic of the Congo (13.2%), the United Republic of Tanzania (4.1%), and Mozambique (3.8%)”.

>> we modified the text

Line 122: ‘Pic of malaria cases shifted to’—needs correction. Probably, the authors mean to say ‘peak of malaria season’

>> we modified the text

Lines 140 & 141: There is nothing like insecticide spreading strategy in malaria. Please check.

>> it is indoor house spreading effectively.  .. we modified

Figure 1: A, C, D: X & Y axis are not clear.

>> sorry for that, the quality of the figure will be improved as TIFF file with the editing version

Figure 2.:  figures are blurred and not clear. It is suggested to replace them with legible figures.

>> sorry for that, the quality of the figure will be improved as TIFF file with the editing version

Ethics approval and consent to participate: It appears data used for this study had approval by the National, ethics committee of Niger for a different purpose. Would the same ethical clearance hold good for this publication also?

>> NEC doesn’t give approaval for publication but for the study and the sampling . So the authorization applied

Round 2

Reviewer 1 Report

Most of the comments are satisfactorily answered, although the discussion around the impact of the COVID pandemic on malaria transmission due to disruption of services is still missing. This COVID impact is well mentioned into the introduction but not in the discussion where it should be. Further, the projected impacts of the climate changes are based on modeled predictions and this must also be highlighted since predictions may not become real. But since this is an "opinion paper", the discussion, this should be well clarified into the discussion.

Minor changes are still needed in the discussion to include the above points, but no further major revision is requested. The conclusion well summarizes the needs for more studies and data collection to anticipate and prepare for the changes. 

Author Response

we added a new paragraph to include these remarks "Data related to malaria collected in Niger during the last years suggest an evolving situation in time and space of the transmission. These changes must be taken into account by the malaria control programs. However, the last period was heavily affected by the COVID 19 pandemic which could have impacted the registration of cases. In the same line, if most of climate models argue for an increase in rainfall in Sahel, the scale of the data generated doesn’t allow a fine prevision of the local impact of these changes and lot of works remain to be done. The objective of this short manuscript was to support the urgent need of new studies in this field"
